# Informer-Based Safety Risk Prediction of Heavy Metals in Rice in China

**DOI:** 10.3390/foods12030542

**Published:** 2023-01-26

**Authors:** Ping Lu, Wei Dong, Tongqiang Jiang, Tianqi Liu, Tianyu Hu, Qingchuan Zhang

**Affiliations:** 1National Engineering Research Centre for Agri-Product Quality Traceability, Beijing Technology and Business University, Beijing 100048, China; 2School of E-Business and Logistics, Beijing Technology and Business University, Beijing 100048, China

**Keywords:** risk assessment, heavy metals, rice, safety risk prediction, Informer

## Abstract

Focused supervision and early warning of heavy metal (HM)-contaminated rice areas can effectively protect people’s livelihood security and maintain social stability. To improve the accuracy of risk prediction, an Informer-based safety risk prediction model for HMs in rice is constructed in this paper. First, based on the national sampling data and residential consumption statistics of rice, we construct a dataset of evaluation indicators that can characterize the level of rice safety risk so as to form a safety risk space. Second, based on the K-medoids clustering algorithm, we classify the rice safety risk space into levels. Finally, we use the Informer neural network model to predict the safety risk indicators of rice in each province so as to predict the safety risk level. This study compares the prediction accuracy of a self-constructed dataset of rice safety risk assessment indicators. The experimental results show that the prediction precision of the method proposed in this paper reaches 99.17%, 91.77%, and 91.33% for low, medium, and high risk levels, respectively. The model provides technical support and a scientific basis for screening the time and area of HM contamination of rice, which needs focus.

## 1. Introduction

With the acceleration of urbanization and industrialization, the process of mining and smelting and chemical production have caused many HMs to enter water and soil, causing serious environmental pollution and making HMs enrich in agricultural soils. HMs accumulate and migrate through the crop planting system [1] and enter the human body through the food chain, and accumulation to a certain extent poses a threat to human health [2]. Studies have shown that the most important route of exposure to HMs in humans is dietary exposure, where HMs are passed through the food chain and accumulate in the human body [3,4]. Rice is a major food crop in China, with more than 65% of the country’s population consuming rice as a staple food [5], and rice has a strong enrichment capacity for many HMs [6], especially cadmium (Cd), chromium (Cr), and arsenic (As), which can be harmful to human health. Cd is metabolized very slowly in the human body and tends to accumulate in the body, which can be harmful [7,8] and cause cancer and cardiovascular diseases [9,10,11]. Chromium is one of the essential trace elements, but excessive intake can cause serious damage to the lungs, kidneys, liver, stomach, skin, and eyes and even cause cancer [12,13]. Arsenic is a metalloid, but its toxicity is similar to that of HMs, and its long-term accumulation in the body can lead to hypertension, diabetes, cardiovascular disease, skin lesions, and premature births and can cause cancer [14,15].

With the increase in demand for rice in the international as well as domestic consumer markets and the increase in safety awareness, rice quality and safety have become an issue of multifaceted concern for the government and consumers. Therefore, strengthening food quality and safety risk monitoring, risk assessment, and early warning analysis of rice quality and safety are conducive to maintaining social stability, safeguarding people’s livelihood security, and improving international economic status.

Sibuar [16] et al. conducted a quantitative analysis and human health risk assessment of HM contamination in rice soils and rice plants in three regions of Perak, Malaysia, and their findings indicated that HM contamination poses a potential health risk to the population and requires control measures. Raja [17] et al. evaluated the health risk of HMs in the groundwater of industrial townships in India, and the high carcinogenic and noncarcinogenic risk values indicated the presence of health risks from lead, cadmium, nickel, and chromium metals in the study area. Luo [18] et al. analyzed the level, source apportionment, and risk assessment of HM contamination in dust storms in key cities in northwest China, and the experimental results showed that both adults and children had the highest carcinogenic and noncarcinogenic risks from the ingestion route, with higher risks in children than in adults. Xiang [19] et al. focused on the relationship between the risk of HM contamination within soil–crop systems, and their study confirmed that HMs in soil can affect HMs in crops to different degrees and that HM interactions are important for contamination risk control. At present, a large number of studies have focused on the current status of HM contamination in rice and soil in various regions of China, but these studies have mainly focused on certain local areas, and relatively few studies have been conducted to systematically and comprehensively assess and analyze the HM contamination and health risks of major crops on a national scale.

In recent years, time series analysis has often been used to reveal the development and change pattern of a phenomenon or to portray the intrinsic quantitative relationship between a phenomenon and other phenomena and its change regularity from a dynamic perspective and has also been widely used in hydrological forecasting [20,21,22], environmental pollution control [23,24], astronomy [25,26], and oceanography [27,28]. This property can also meet the needs of food safety risk prediction. Jiang [29] et al. used a Transformer deep learning algorithm to predict the safety risk level of antibiotic residues in freshwater products in China. Wang [30] et al. used a novel long- and short-term memory neural network to integrate a sum-product-based analytic hierarchy process for risk warning of food safety. Jiang [31] et al. used a deep learning model to grade and predict the safety risk of pesticide residues in vegetables.

Based on the comprehensive review above, this paper constructs an Informer-based safety risk prediction model of HMs in rice in China. Using the data on the content of three HMs in rice from the food safety sampling samples of the State Administration of Market Supervision and Administration, combined with the rice consumption data of Chinese residents, the rice safety risk assessment indicator dataset is constructed, and the K-medoids algorithm is combined to establish the rice safety risk rank space. Based on the Informer neural network model, the safety risk assessment indicators of rice in each province are predicted based on historical data on the constructed dataset, and the predicted indicator values are used to determine the risk level of rice in that province at that time. The model proposed in this paper provides technical support and a scientific basis for screening the time and area of HM contamination of rice, which needs focus.

## 2. Materials and Methods

### 2.1. Materials

#### 2.1.1. Data

The sampling data of HM content in rice in this study are obtained from the 2019–2021 sampling samples from the State Administration of Market Supervision and Administration involving 20 provinces with a total of 180,368 samples, where the As sampled in the samples is inorganic arsenic. The limits of HMs in Chinese rice are specified in the National Standard for Food Safety Limits of Contaminants in Food, with limits of 0.2 mg/kg for Cd, 1.0 mg/kg for Cr, and 0.2 mg/kg for As (inorganic).

The FAO/WHO Expert Committee on Food Additives (JECFA) and U.S. Environmental Protection Agency (EPA) have also jointly made provisions for the average daily reference dose (RfD) of HMs, in which the RfD values for Cd, Cr, and As are 0.001, 0.003, and 0.0003 mg/(kg·d), respectively. In addition, the EPA’s cancer slope factor (CSF) for HMs is also made clear, in which the CSF values for Cd, Cr, and As are 6.3, 0.5, and 1.5 kg·d/mg, respectively.

The rice consumption data in this study are obtained from the Fifth Chinese Total Diet Study [32], which used a questionnaire combined with a multistage stratified random sampling method to estimate the main dietary consumption of residents in 20 provinces.

#### 2.1.2. Experimental Environment

In this study, the deep learning framework PyTorch [33] is used to build the network model, and the experimental environment and parameter settings are shown in Table 1 below.

### 2.2. Rice Safety Risk Assessment Model

In order to scientifically carry out the risk classification evaluation of food contaminants based on the main effects caused by food contamination, taking into account the evaluation object and the use of the model in this paper, this paper selects the Nemerow integrated pollution index (NIPI), target hazard quotient (THQ), and total carcinogenic risk (TCR) as the three assessment indicators in the risk assessment model.

In this paper, the sample data of undetected HMs are much less than 60% of the total samples. In accordance with the principle of credible evaluation of low-level contaminants in food proposed by the World Health Organization, the undetected data are given a 1/2 limit of detection (LOD) value for calculation in this paper.

#### 2.2.1. Nemerow Integrated Pollution Index

The NIPI reflects the impact and effect of contaminants on food, taking into account the highest value of the contamination indicator and the average value, highlighting the impact and effect of the contaminants with the largest contamination indicator on food quality and overcoming the defects of the average method of individual contaminant sharing.

NIPI can reflect the degree of food contamination comprehensively, taking into account the highest and average values of the pollution indicator, highlighting the impact and effect of the pollutant with the largest pollution indicator on food quality and overcoming the shortcomings of the average method of individual pollutant sharing, and it is often used as a weighted multifactor environmental quality indicator to assess the atmosphere [34,35], water quality [36,37], soil [38,39,40], etc. In this paper, we use NIPI to calculate the integrated contamination level of HMs in the sampled rice by combining the rice sampling data of each province. Generally, the integrated pollution indicator is less than or equal to 1, which means that the rice is not contaminated, and greater than 1 means that the rice is contaminated; the greater the value of the integrated pollution indicator, the more serious the contamination. The expression of the single-factor pollution indicator of HMs is shown below:(1)$$P_{i,j}=\frac{C_{i,j}}{S_{j}}$$
where $P_{i,j}$ is the single-factor contamination indicator of HMs in rice in province i; $C_{i,\;j}$ is the detection value of j-HM in rice in province i (mg/kg); and $S_{j}$ is the national limit of j-HM in rice (mg/kg).
(2)$$PI_{i}=\sqrt{\frac{P_{max(i)}^{2}+P_{ave\left(i\right)}^{2}}{2}}$$
where $PI_{i}$ is the Nemerow integrated pollution indicator of rice in province i; $P_{max(i)}$ is the maximum value of the single-factor pollution indicator of HMs in rice in province i; and $P_{ave\left(i\right)}$ is the average value of the single-factor pollution indicator of HMs in rice in province i.

#### 2.2.2. Target Hazard Quotient

In this paper, noncarcinogenic risks are evaluated using the target hazard quotient proposed by the United States Environmental Protection Agency (USEPA) [41,42] to assess the cumulative health risks of HMs to humans using the health risk assessment method for human exposure to contaminants. Assuming that the toxic effects of each HM on humans are cumulative and there is no synergistic or antagonistic relationship [43], the compound health risk, *THQ*, of the inhabitants consuming three HMs in rice can be calculated as the sum of the cumulative single metal hazard factors, *HQ*, for the three metals. If THQ≤1, the noncarcinogenic risk is considered low, and if *THQ*
> 1, the noncarcinogenic risk is considered to exist. The expressions are shown below:(3)$$HQ_{i,j}=\frac{F_{i}^{50}\times C_{avg\left(i,j\right)}}{RfD\times W}$$
where $HQ_{i,j}$ is the noncarcinogenic risk factor of class j HM in rice in province i; $F_{i}^{50}$ is the average consumption of rice in province i (kg/d); $C_{avg\left(i,j\right)}$ is the average content of class j HMs in rice in province i (mg/kg); W is the average human body weight (kg), taken as 60 kg; and *RfD* is the average daily reference dose as described above.
(4)$$THQ_{i}=\underset{j}{\sum }HQ_{i,j}$$
where ${THQ}_{i}$ is the target hazard quotient of rice in province i, which is the cumulative sum of the single metal noncarcinogenic risk factors, *HQ*, for the three HMs.

#### 2.2.3. Total Carcinogenic Risk

TCR is commonly used to calculate the lifetime probability of developing any type of cancer in an individual due to exposure to carcinogens [44,45,46,47], and in this paper, TCR is used to assess the carcinogenic risk of HMs in rice as the product of contaminant exposure dose and carcinogenic intensity indicator to reflect the likelihood of long-term carcinogenic risk occurrence. The expression of the carcinogenic risk assessment equation is as follows:(5)$$EDI_{i,j}^{50}=\frac{F_{i}^{50}\times C_{avg\left(i,j\right)}}{W}$$
where $EDI_{i,j}^{50}$ is the average daily intake of HM j through rice per kilogram of body weight in the population of province i (mg/kg·d), and the meanings of $F_{i}^{50}$, $C_{avg\left(i,j\right)}$, and W are as described above.
(6)$$TCR_{i}=\underset{j}{\sum }\frac{EF\times ED\times CSF_{j}\times EDI_{i,j}^{50}}{AT_{C}}$$
where $TCR_{i}$ is the total carcinogenic risk of multiple HMs in province i; EF is the exposure frequency (365 days/year); ED is the exposure duration (70 years); $CSF_{j}$ is the carcinogenic intensity indicator of HM j (kg·d/mg); and $AT_{C}$ is the duration of carcinogenic effect (365 days/year × exposure duration, assumed to be 70 years in this paper).

### 2.3. Risk Classification Model Based on K-Medoids

Based on the rice safety risk assessment model constructed above, this paper constructs a dataset of weekly rice risk assessment indicators for 20 provinces nationwide, based on which a three-dimensional space of rice safety risk assessment indicators is constructed. In order to grade the rice safety risk more objectively, this paper adopts an unsupervised clustering algorithm to automatically grade the rice safety risk assessment indicator data constructed above so as to construct a grade space for the risk. Since the distribution of data points in the data space of different levels is more scattered, in order to reduce the sensitivity of the algorithm to isolated points, this paper uses the K-medoids algorithm [48] to cluster the data, and the K-medoids algorithm selects the object closest to the mean in the cluster as the cluster center. The algorithm is specified as follows.

(1)Randomly select k representative objects as the initial centroids.(2)Assign each remaining object to the cluster represented by the nearest centroid.(3)Randomly select a noncentroid object y.(4)Calculate the total cost f of replacing the centroid x with y.(5)If f is negative, then replace x with y to form a new centroid.(6)Repeat (2)–(5) until k centroids no longer change.

### 2.4. Prediction Model of Rice Safety Risk Level Based on Informer

#### 2.4.1. Prediction Model of Rice Safety Risk Level

Deep learning algorithms are widely used in medical diagnosis, geological exploration, and industrial equipment fault diagnosis; however, in time series prediction problems, the distribution of the series may change continuously as the time axis advances, which requires models with stronger extrapolation capabilities. Therefore, in order to predict the safety risk level of rice, a rice safety risk level prediction model based on Informer [49] is constructed in this paper, which consists of three layers of structure, namely, a data layer, risk indicator prediction layer, and risk level forecast layer, and the model structure is shown in Figure 1.

In the data layer, after preprocessing the rice sampling data from 2019 to 2021, such as cleaning and integration, and combining them with the rice consumption data of the national residents, the weekly rice safety risk assessment indicators of 20 provinces over 3 years are calculated, including NIPI, THQ, and TCR, based on which the rice safety risk assessment indicator dataset is constructed and the spatial construction of the rice safety risk sample points is completed. The time series of each risk assessment indicator ${[X}_{t}{,\ldots ,X}_{t+i}{,\;\ldots X}_{T}]$ is used as input into the risk indicator prediction layer to predict the time series of each risk assessment indicator X(NIPI), X(THQ) and X(TCR) at the time of [T + 1, …, T + k].

In the risk indicator prediction layer, this paper uses the Informer algorithm to predict three safety risk assessment indicators of HMs in rice with time series characteristics. The Informer model completes the construction of encoder and decoder through a multihead ProbSparse self-attention layer, multihead attention mechanism, and masked multihead ProbSparse self-attention layer to solve the long-term temporal prediction problem and improve the performance of the prediction model. A detailed description of the Informer-based predictive model for rice safety risk assessment indicators is given in Section 2.4.2.

Finally, in the risk level forecast layer, the time series [T + 1, …, T + k] of each safety risk assessment indicator is output by the risk indicator prediction layer, the Euclidean distance between the predicted safety risk assessment indicator and the risk safety level space that has been divided is measured, and the rice risk level of that week in that province is divided into the level space with the closest distance.

#### 2.4.2. Predictive Model of Rice Safety Risk Assessment Indicator Based on Informer

The food safety risk assessment model constructed in this paper contains rich information on the degree of rice contamination, and the prediction of rice safety risk level can be achieved by predicting the rice safety risk assessment indicator. Informer is an improved network model of Transformer, which reduces the time complexity of the network model by assigning greater weights to the important features using a multiheaded probabilistic sparse self-attention mechanism while improving the prediction speed based on generative decoding and solving the problem of long-time dependence of time series data [49]; therefore, in this paper, we use the Informer neural network model to predict food safety risk assessment indicators, as shown in Figure 2, and the model consists of two parts: encoder and decoder.

The time series ${[X}_{t}{,\ldots ,X}_{t+i}{,\ldots X}_{T}]$ of the constructed rice safety risk assessment indicator is used as the input of the Informer network, and the temporal correlation of the time series ${[X}_{t}{,\ldots ,X}_{t+i}{\ldots X}_{T}]$ is fully exploited by using location encoding, i.e., local timestamps and global timestamps to mark the local and global backward and forward temporal location relationships of the time series. The multiheaded attention mechanism of Informer is used to focus attention on the more obvious data features of contamination to obtain the long-term dependence of the time series of rice safety risk assessment indicators. The decoder input consists of two parts, one of which is the implied intermediate feature data about the rice safety risk assessment indicator output by the encoder and the other which needs to place the rice safety risk assessment indicator such that it is predicted at the input using 0 to occupy it and add a masking mechanism to prevent each location from focusing on the rice safety risk assessment indicator information at future time points. The data are connected to a multiheaded attention mechanism, which then connects a fully connected layer to output rice safety risk assessment indicator prediction data $X\left(\mathrm{NIPI}\right),X\left(\mathrm{THQ}\right)$ and X(TCR), thus realizing the prediction of rice safety risk level.

The attention mechanism is calculated as shown below, which contains 3 vectors, query vector ($Q\in R^{L_{Q}\times d}$), key vector ($K\in R^{L_{X}\times d}$) and value vector ($V\in R^{L_{V}\times d}$). Let $q_{i},k_{i}$, and $v_{i}$ be the i−th row of *Q*, *K*, and *V*, respectively; then, the attention of the i−th query is defined as a kernel smoother in the form of probability.
(7)$$A\left(q_{i},K,V\right)=\underset{j}{\sum }\frac{k(q_{i},k_{j})}{\sum_{l}k\left(q_{i},k_{l}\right)}v_{j}=E_{p(k_{j}|q_{i})\left[V_{j}\right]}$$

The sparsity measure of the i−th query is given by:(8)$$M\left(q_{i},K\right)=ln\overset{L_{k}}{\underset{j=1}{\sum }}e^{\frac{q_{i}k_{j}{}^{T}}{\sqrt{d}}}-\frac{1}{L_{k}}\overset{L_{k}}{\underset{j=1}{\sum }}\frac{q_{i}k_{j}{}^{T}}{\sqrt{d}}$$

Based on the proposed metric, ProbSparse self-attention is obtained by allowing each key to focus on only u primary queries, as shown in the following equation, where $\bar{Q}$ is the sparse matrix of the first u Query.
(9)$$Attention\left(Q,K,V\right)=softmax\left(\frac{\bar{Q}K^{t}}{\sqrt{d}}\right)V$$

The Informer encoder is responsible for encoding the input time series of rice safety risk assessment indicators, obtaining the time series dependencies of rice safety risk assessment indicators, and mapping them into intermediate features containing information on rice safety risk assessment indicators, and its internal structure consists of two stacks with the same operation. The stack structure is shown in Figure 3. Stack1 is the main stack and receives the entire input sequence, while Stack2 takes half of the input slices. Each stack consists of an encoding layer and a distillation layer, which includes a multiheaded probabilistic sparse self-attentive layer, a forward neural network, residual connectivity, and regularization operations, as shown in Equation (10). The distillation layer improves the robustness of the network and reduces the memory used by the network through the distillation mechanism. The output of all stacks is concatenated to obtain the final hidden representation of encoder, i.e., the feature map shown in Figure 2.
(10)$$O=LayerNorm\left(x+Sublayer\left(x\right)\right)$$
where Sublayer is a multiheaded sparse self-attentive mechanism and forward neural network processing function and LayerNorm is the regularization function.

The distillation mechanism uses a one-dimensional convolution operation in the time dimension and halves the input length by adding a pooling layer after the ELU activation function, where the distillation layer is one layer less than the encoding layer. The distillation operation advances from the j−th layer to the $\left(j+1\right)-\mathrm{th}$ layer as:(11)$$x_{j+1}=MaxPool\left(ELU\left(Conv1d\left({\left[x_{j}\right]}_{AB}\right)\right)\right)$$
where $\left[\cdot \right]$ denotes the key operation including the multihead probabilistic sparse self-attentive operation and the attention block, Conv1d denotes the one-dimensional convolution operation, MaxPool is the maximum pooling operation, and ELU is the activation function, which is calculated as:(12)$$ELU\left(x\right)=\left\{\begin{array}{c} e^{x}-1,\;x<0\\ x,\;x\geq 0 \end{array}\right.\;$$

## 3. Results and Discussion

### 3.1. Performance Evaluation Metrics

The safety risk level of rice is decided by the position of the spatial points constructed by the above three indicators in the risk level space; therefore, the goodness of the prediction of the indicators directly determines the accuracy of the risk level classification of rice. This paper separately examines the performance of risk assessment indicator predictions, and the performance of the model in predicting risk levels based on risk indicators is evaluated.

#### 3.1.1. Performance Evaluation of Indicator Prediction

Root mean square error (*RMSE*) and mean absolute error (*MAE*) are chosen in this study to evaluate the efficacy of the models proposed in the paper to predict the single risk assessment metrics of NIPI, THQ, and TCR. The equations for these two evaluation metrics are shown below.
(13)$$RMSE=\sqrt{\frac{1}{n}\overset{n}{\underset{i=1}{\sum }}{\left(y_{i}-{\hat{y}}_{i}\right)}^{2}}$$
(14)$$MAE=\frac{1}{n}\overset{n}{\underset{i=1}{\sum }}\left|y_{i}-{\hat{y}}_{i}\right|$$
where $y_{i}$ denotes the value of a risk assessment indicator of rice in week i calculated from the national sampling data and residential consumption data and ${\hat{y}}_{i}$ denotes the value of a risk assessment indicator of rice in week i predicted from historical data.

#### 3.1.2. Performance Evaluation of Risk Level Prediction Model

In this paper, the performance of each risk level prediction is evaluated separately using three evaluation metrics: precision, recall rate, and F1 value, as shown in Equations (15)–(17).
(15)$$Pecision=\frac{TP}{TP+FP}$$
(16)$$Recall=\frac{TP}{TP+FN}$$
(17)$$F1=\frac{2*P*R}{P+R}$$
where TP represents the number of samples whose risk level is correctly predicted by the model, FP represents the number of samples whose risk level is predicted by the model but it is an incorrect risk level, and FN represents the number of samples whose risk level is predicted by the model to be other risk levels but not the correct risk level. To be compatible with precision and recall, F1 values are used to evaluate the predictive performance of the model in a comprehensive manner.

### 3.2. Rice Safety Risk Grading

#### 3.2.1. Dataset of Rice Safety Risk Assessment Indicators

Based on the rice safety risk assessment model constructed above, the dataset of rice safety risk assessment indicators from 2019 to 2021 is obtained using hierarchical analysis, including three safety risk indicators, NIPI, THQ, and TCR, for each week for 20 provinces. The datasets are shown in Figure 4, Figure 5 and Figure 6.

#### 3.2.2. Safety Risk Classification

As seen from Figure 4, Figure 5 and Figure 6, the rice safety risk assessment indicators differ greatly in order of magnitude; therefore, in order to eliminate the influence of magnitudes between indicators, data standardization is required to address the comparability between data indicators. Therefore, in this paper, the maximum–minimum normalization is applied to the safety risk assessment indicators to eliminate the undesirable effects caused by odd sample data, as shown in Equation (18), where $index^{\prime }$ denotes the normalized indicator, index denotes the original indicator, $index_{min}$ denotes the minimum value in the assessment indicator, and $index_{max}$ denotes the maximum value in the assessment indicator.
(18)$$index^{\prime }=\frac{index-index_{min}}{index_{max}-index_{min}}$$

In this study, the K-medoids algorithm is used to cluster the above-constructed dataset of risk assessment indicators, and the three indicators of NIPI, THQ, and TCR of rice are used as the feature vectors. The number of different clusters selected for the data is the number selected to classify the constructed safety risk space into levels. In this paper, the silhouette coefficient is used to measure whether the selection of the number of clusters is reasonable and effective, where the larger the value of the silhouette coefficient, the better the clustering effect. The number of clusters is selected from two to seven in the experiment, and the silhouette coefficients of clustering are shown in Figure 7.

It can be seen from Figure 7 that the silhouette coefficient is the largest when the number of clusters is chosen as three, indicating that when the number of clusters is chosen as three, the sample points within the clusters are compact among each other, while the distance between clusters is large. Therefore, this experiment divides the safety risk assessment levels into three levels, and the cluster centers in each level after data normalization and the number of samples in each level are shown in Table 2. The risk levels are divided according to the distance between the cluster centers and the origin, and it can be seen from Table 2 that as the risk level increases, the assessment indexes also increase.

#### 3.2.3. Analysis of Grading Results

According to the above grading results, the distribution of each risk assessment indicator of the points in the space of different safety risk levels that have been classified is statistically shown in Figure 8, Figure 9 and Figure 10, from which it can be seen that
(1)Category 1 is a cluster of low-risk points, which is characterized by the fact that the distributions of NIPI, THQ, and TCR are all concentrated in a small range, concentrated in ranges of 0–0.0085, 0–0.004, and 0–0.015 × 10^−5^, respectively, and the numerical magnitudes are also small, but most of the data are concentrated in cluster 1.(2)Category 2 is a cluster of medium-risk points, which is characterized by a larger interval of NIPI, THQ, and TCR relative to Category 1, with distribution intervals concentrated in ranges of 0.1–0.75, 0.1–0.75, and 1 × 10^−5^–3.5 × 10^−5^, respectively.(3)Category 3 is a cluster of high-risk points, which is characterized by the data values of NIPI, THQ, and TCR being much larger than those in Category 1 and Category 2, and the intervals are distributed in ranges of 1–1.5, 0.5–0.8, and 0.00002–0.00005, respectively.

**Figure 8 foods-12-00542-f008:**
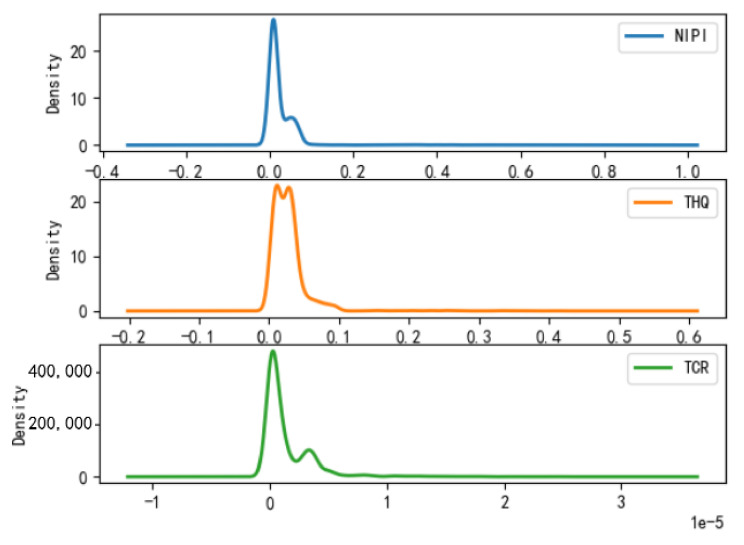
Plot of probability density function of Category 1.

**Figure 9 foods-12-00542-f009:**
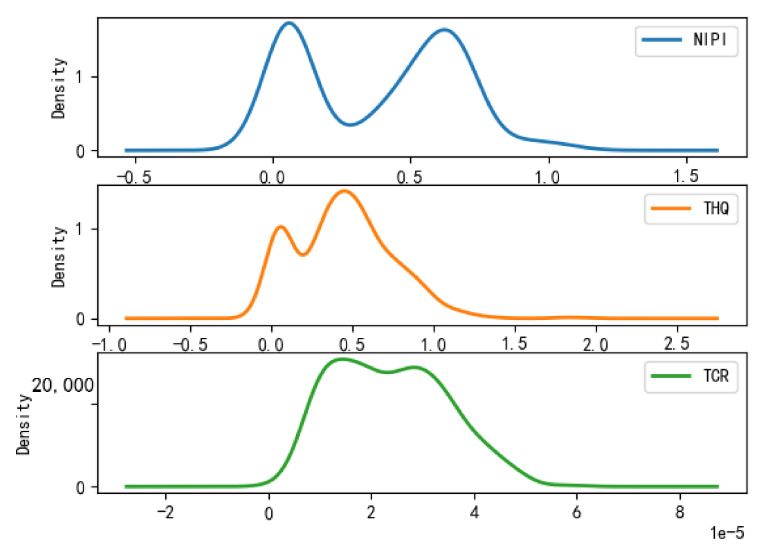
Plot of probability density function of Category 2.

**Figure 10 foods-12-00542-f010:**
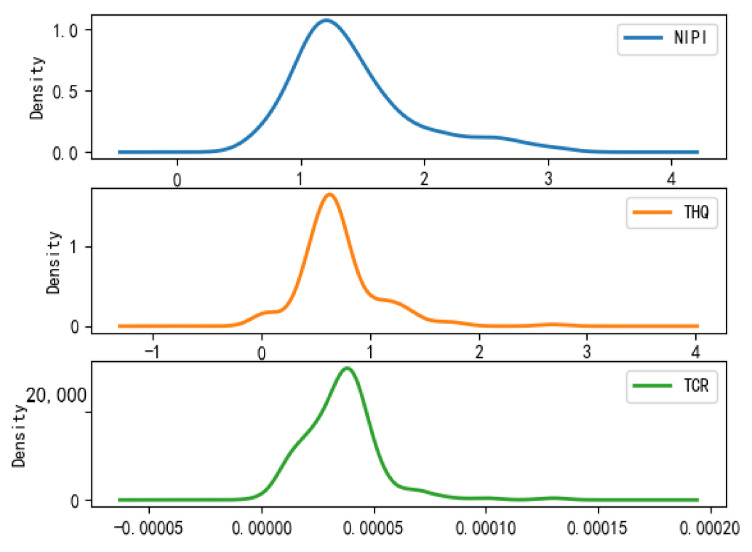
Plot of probability density function of Category 3.

From the above analysis, it can be seen that the value distribution interval in Category 1 is relatively the smallest and that the overall data value is also the smallest, but the number is the largest, which represents rice with a small risk hazard level. The indicators in Category 2 are at a medium level, so the sample points represented are rice at a medium risk level and the quantity is also at a medium level. Category 3 has the overall maximum value of each assessment indicator but the least amount, which represents rice at a high risk level, requiring focused supervision of the corresponding sample points.

### 3.3. Rice Safety Risk Level Prediction

Based on the model described above, this experiment first predicts the rice safety risk assessment indicator. The prediction results are shown in Figure 11, Figure 12, Figure 13, Figure 14 and Figure 15, where the solid blue line represents the actual data trend, the solid red line represents the data used for the training fit, and the dashed red line represents the predicted data trend. Weeks 1–138 in this experiment are the training set of the experiment, and weeks 139–159 are the test set of the experiment.

In addition, to demonstrate the effectiveness of the present model, LSTM, GRU, and Transformer prediction models are used in this paper for comparison experiments with the model proposed in this paper. Figure 16 and Figure 17 show the metrics RMSE and MAE for the evaluation of the prediction effectiveness of each model.

From Figure 11, Figure 12, Figure 13, Figure 14 and Figure 15, it can be seen that the level of rice contamination is low in most areas of China; however, rice is more seriously contaminated with HMs in some areas at certain times of the year and requires focused government supervision.

The provinces with serious pollution are mainly concentrated in Fujian, Guangxi, Hunan, Jiangxi, Sichuan, and some other cities with rich water resources. These provinces with more serious contamination mainly have two to three indicators in the risk assessment model that are high, such as Sichuan Province, where three risk assessment indicators are high for a longer period of time, which means that the NIPI, THQ, and TCR values of rice in this province are large and the HM contamination of rice in this province is more serious and has obvious health hazards and carcinogenic risks to the exposed population.

As seen from the graph, the weeks with more serious HM contamination of rice are concentrated at about 24–44 weeks, 77–97 weeks, and 130–150 weeks, which means that the contamination is mainly concentrated from June to November every year, and these months are the period when rice is newly launched in China every year. Due to the vast territory of China, the climatic environment of each production area varies, so there is a certain difference in the time of the new rice market. The northern region has a yearly season, mainly concentrated around September on the market, but some areas are in late October, such as the northeast new rice marketed from about late October to early November, while the southern region is mostly a yearly two-season rice, concentrated in July–November on the market. Therefore, the government needs to focus on the relevant provinces and cities when new rice is available from June to November.

In addition, it is also found from the graph that the predicted trend of certain indicators is different from the actual trend; for example, the TCR indicator in Zhejiang Province decreases in December 2021, while the predicted value is still at a high level, and it is found through the research that the local government helps to reduce the HM pollution of crops by vigorously managing water resources.

However, there is no shortage of cities where the actual indicators suddenly rose, such as Hunan Province, where all three indicators are suddenly at high levels at week 155. The research has found that the province and city were increasing urban construction in 2021, resulting in the pollution of water resources, which in turn led to HM contamination of crops, so the new rice marketed in November was sampled from more substandard products.

From the figure, it is easy to find that the trends of the three indicator values are not always consistent; for example, the indicators of NIPI and THQ in Zhejiang Province generally show small and stable values, while the indicator of TCR generally shows high values, which shows that the model assesses the safety risk of rice from multiple dimensions and gives a risk level for the safety risk of rice by combining all risk items.

Figure 16 and Figure 17 show the statistical results of the RMSE and MAE evaluation metrics of the four neural-network-based prediction models, respectively. Larger values of the RMSE and MAE assessment metrics indicate larger errors in forecasting. From the figure, it can be seen that the prediction effect of the Informer-based prediction model proposed in this paper is better than the other models for all three indicators. The RMSE and MAE of the TCR indicator are much smaller than the other indicators because the TCR itself is of a small order of magnitude, so the error value is also smaller, but it can be seen from the experimental results that the prediction of the Informer-based neural network model is still better than the other models.

After predicting the rice safety risk indicator for each week in each province using the above four neural network models, the distance between the sample point constructed by this indicator and the already divided cluster centers is calculated, and the sample point is grouped into the nearest cluster, which determines the risk level of the sample point. The precision, recall rate, and F1 values of the predicted low, medium, and high risk levels are calculated for each of the four models, as shown in Table 3. The experimental results show that the number of sample points at low, medium, and high risk levels differ greatly from each other, and therefore, the calculated evaluation metrics at each risk level also differ greatly from each other. Low levels have the most sample points and thus the best performance of the calculated evaluation indicators, while high risks have fewer sample points and the worst performance of the calculated indicators. By comparing the performance of the four models at the same risk level, it is found that the prediction effect of Informer is much higher than other prediction models. From the medium- and high-risk prediction results, we can see that the F1 value of the prediction model proposed in this paper reaches 93.09% and 90.13%, respectively, which can meet the current demand of early warning.

## 4. Conclusions

Food safety problems caused by HMs and metal-like contamination are gradually increasing, and the damage caused by HM poisoning to an organism is multisystem, multiorgan, and irreversible. HMs are enriched and difficult to decompose in the environment and can be a direct threat to human health. Given China’s vast geography and rich production, not only the lack of management resources and manpower of the relevant departments but also the lack of key supervision of key areas can lead to frequent food safety incidents. From the above experiments, it can be seen that medium- and high-risk rice is, after all, a minority in the dataset; therefore, it is necessary to focus on monitoring and early warning in areas where rice contamination is more serious and at the relevant time, which can strengthen food safety risk prevention and control and reduce various adverse effects caused by food safety problems in order to better cope with food safety crises. However, we also see from the experiments that the prediction of this model is based on historical data, so it is not very effective in predicting data after government regulatory interventions or after pollution caused by urban construction, and we also need enough historical data to ensure the training of the model. In addition, as industrialization intensifies, HM mercury contamination has gradually become a global concern. HM mercury is able to circulate in various forms through the atmosphere, water, and soil to different parts of the world. Because of this natural cycle, the release of mercury from any part of the world may affect a completely different part of the world and also affect the predicted results in the experiment, so we will add the study of HM mercury pollution to our next work.

## Figures and Tables

**Figure 1 foods-12-00542-f001:**
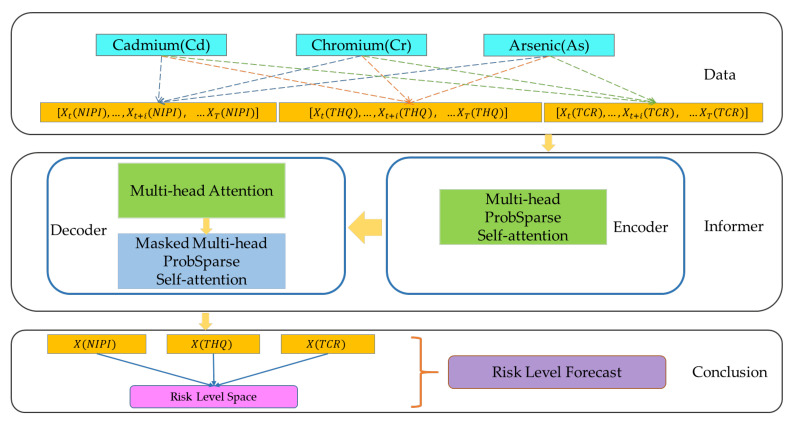
Prediction model of rice safety risk level.

**Figure 2 foods-12-00542-f002:**
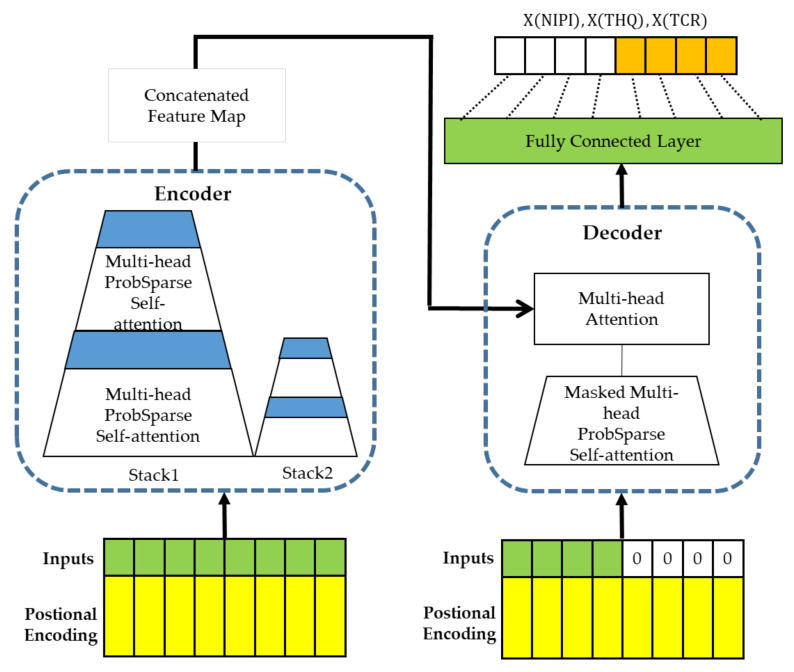
Predictive model of rice safety risk assessment indicator based on Informer.

**Figure 3 foods-12-00542-f003:**
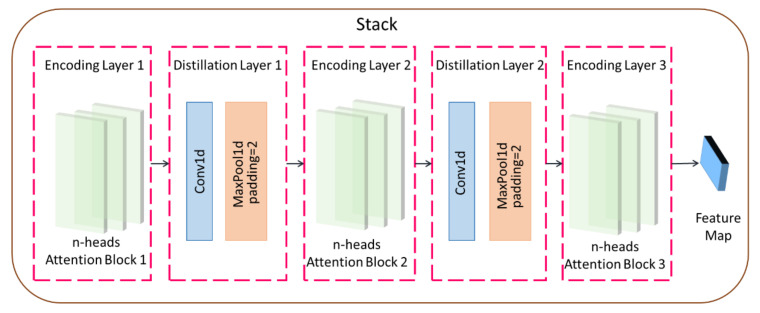
Stack structure diagram.

**Figure 4 foods-12-00542-f004:**
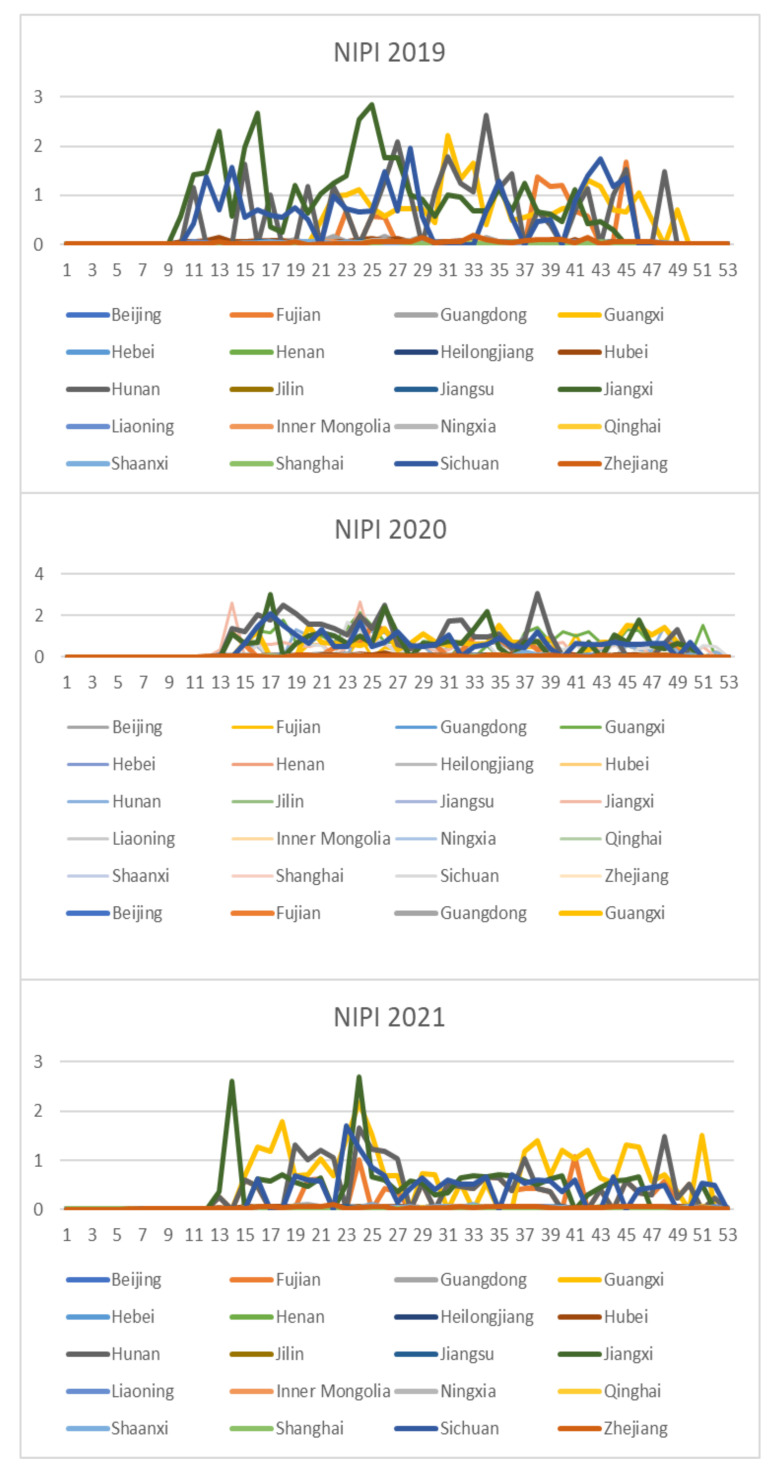
Dataset of rice safety risk assessment indicator NIPI.

**Figure 5 foods-12-00542-f005:**
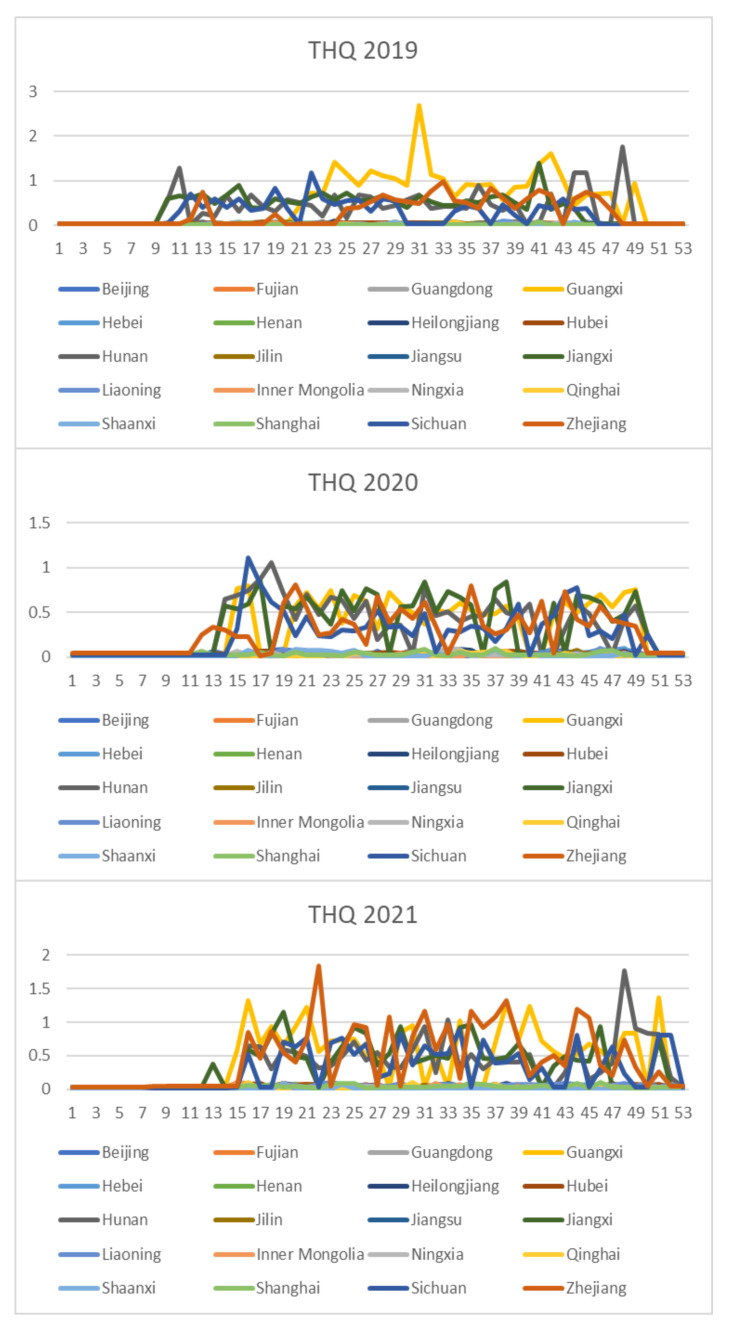
Dataset of rice safety risk assessment indicator THQ.

**Figure 6 foods-12-00542-f006:**
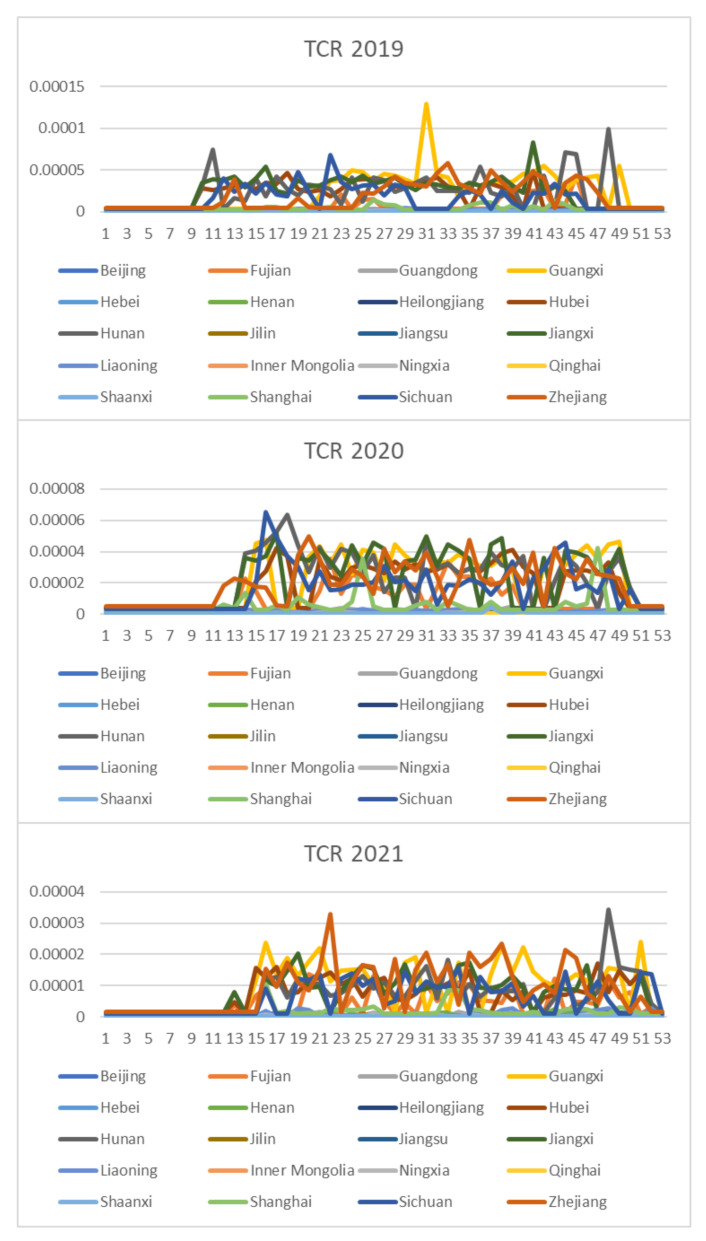
Dataset of rice safety risk assessment indicator TCR.

**Figure 7 foods-12-00542-f007:**
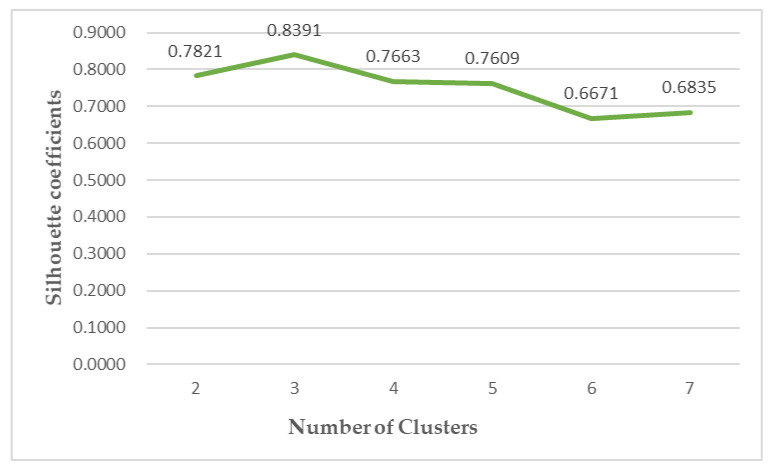
Silhouette coefficients of clusters.

**Figure 11 foods-12-00542-f011:**
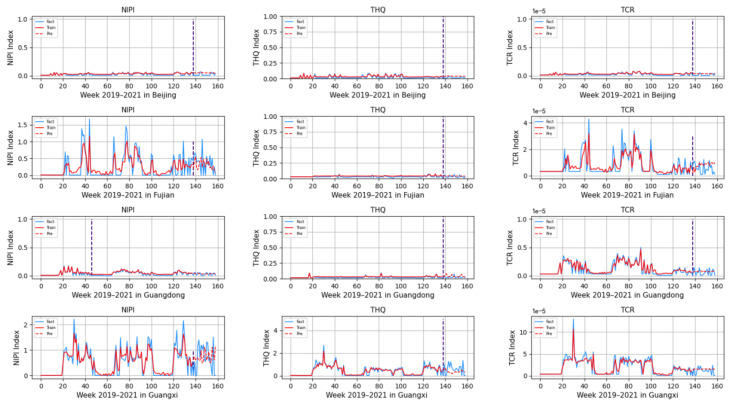
Predicted NIPI, THQ, and TCR indicators for Beijing, Fujian, Guangdong, and Guangxi.

**Figure 12 foods-12-00542-f012:**
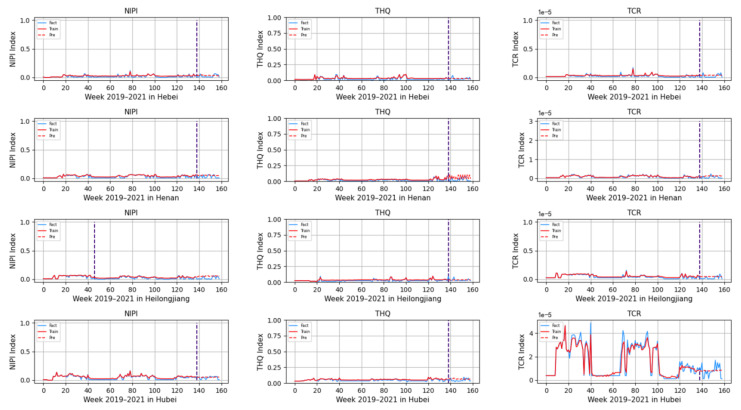
Predicted NIPI, THQ, and TCR indicators for Hebei, Henan, Heilongjiang, and Hubei.

**Figure 13 foods-12-00542-f013:**
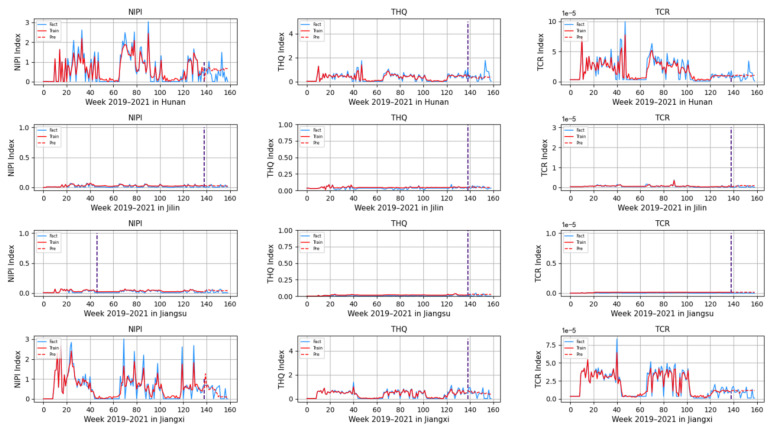
Predicted NIPI, THQ, and TCR indicators for Hunan, Jilin, Jiangsu, and Jiangxi.

**Figure 14 foods-12-00542-f014:**
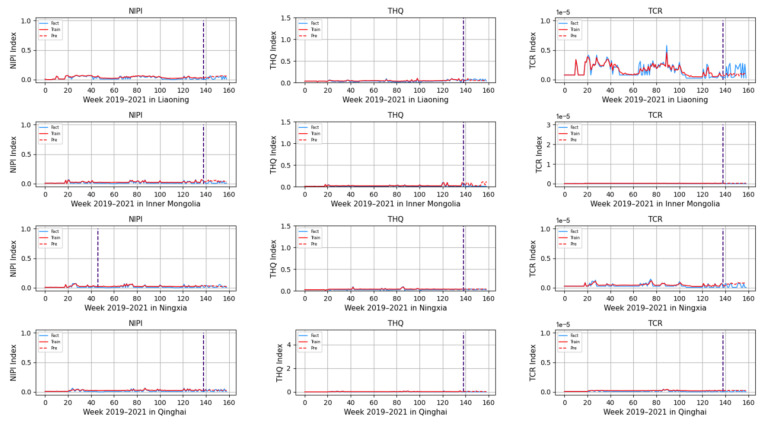
Predicted NIPI, THQ, and TCR indicators for Liaoning, Inner Mongolia, Ningxia, and Qinghai.

**Figure 15 foods-12-00542-f015:**
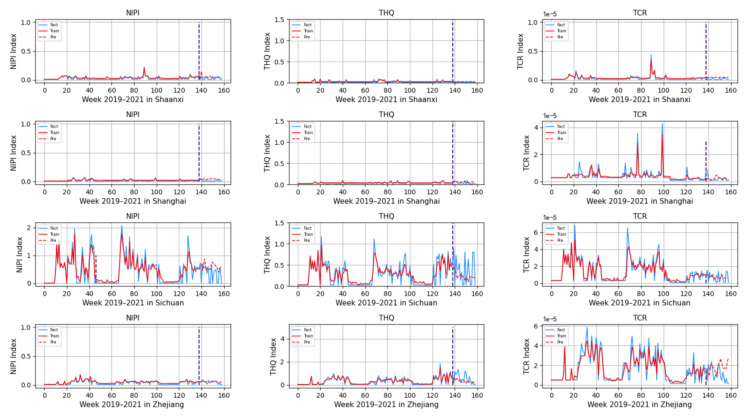
Predicted NIPI, THQ, and TCR indicators for Shaanxi, Shanghai, Sichuan, and Zhejiang.

**Figure 16 foods-12-00542-f016:**
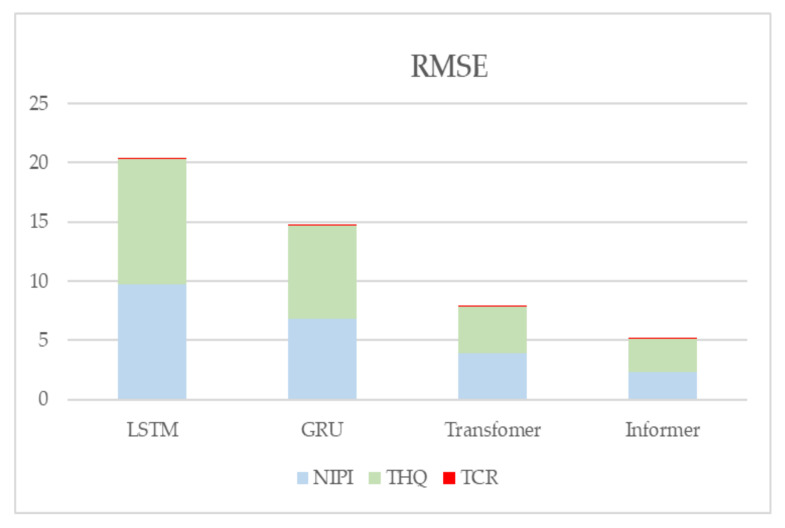
Predictive evaluation metric RMSE for NIPI, THQ, and TCR.

**Figure 17 foods-12-00542-f017:**
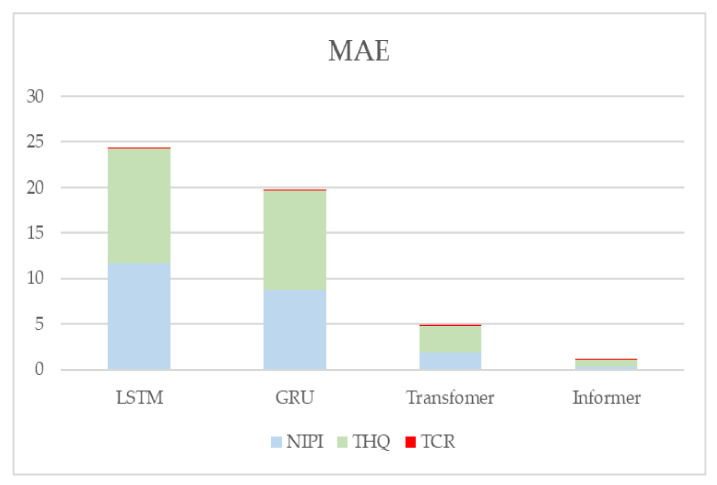
Predictive evaluation metric MAE for NIPI, THQ, and TCR.

**Table 1 foods-12-00542-t001:** Experimental platform and environmental parameters.

Computer Information	Operating System	Windows 10 64-bit
	CPU	Intel(R) Core(TM) i5-8265U CPU @ 1.60 GHz (8 CPUs) ~1.8 GHz
	GPU	Radeon 540X Series
	RAM	16 GB
Toolkit	Python 3.6	Numpy 1.19.4
		Scikit_Learn 0.21.3
		Pandas 0.25.1
		Torch 1.8.0
		Matplotlib 3.1.1

**Table 2 foods-12-00542-t002:** Clustering center and ranking.

Category	NIPI	THQ	TCR	Sample Size	Risk Level
1	0.008093	0.009668	0.010973	2648	Low
2	0.126264	0.164193	0.18236	378	Medium
3	0.464388	0.263196	0.276102	154	High

**Table 3 foods-12-00542-t003:** Risk level prediction evaluation criteria.

Model	Low Level			Medium Level			High Level		
	P%	R%	F1%	P%	R%	F1%	P%	R%	F1%
LSTM	98.21	97.21	97.70	83.76	87.30	85.49	69.70	74.68	72.10
GRU	98.48	97.70	98.09	85.97	89.15	87.53	74.53	77.92	76.19
Transformer	98.78	98.19	98.48	87.95	90.74	89.32	79.75	81.82	80.77
Informer	99.17	98.94	99.06	91.77	94.44	93.09	91.33	88.96	90.13

## Data Availability

Restrictions apply to the availability of these data. Data were obtained from the State Administration for Market Regulation Statistics and are available at [50] with the permission of the State Administration for Market Regulation Statistics.

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
