# Peer review of "Informer-Based Safety Risk Prediction of Heavy Metals in Rice in China"

_foods, 2023, doi:10.3390/foods12030542_

Round 1
Reviewer 1 Report
The research reported in the manuscript entitled “Informer-based Safety Risk Prediction of Heavy Metals in Rice in China” looks interesting and innovative. Some minor comments are given below.
Reviewer report
The research reported in the manuscript entitled “Informer-based Safety Risk Prediction of Heavy Metals in Rice in China” looks interesting and innovative. Some minor comments are given below which needs to be considered for possible publication in the International Journal like “Foods”.
1. In the abstract section, the results about higher prediction accuracy of the model should be given in one sentence with data (how much accuracy).
2. The objectives of the study should be clearly given in the last paragraph of the Introduction section. The objectives of the study given in the last paragraphs are not clear.
3. It will be better the section “Data Set and Experimental Environment” under the Result section should be moved to methodology section.
4. Presentation of the results is good. However, the discussion section needs to be improved in line that results found in this study needs to be discussed with a link to the previous study with citations.
5. Conclusions of the study are not given. It should be included in the manuscript.
The manuscript can be accepted after the above minor revision in the journal like Foods.
Reviewer 2 Report
The preparation and write-up of the manuscript are very well, although a couple of technical suggestions should be followed-
1. The figure numbers are too high and some of the figures need proper result analysis. Figure 7, for example, is not discussed in the manuscript.
2. The discussion is too short for such a large data set and results. Please elaborate more on this.
3. The modellings are well presented and described but the connections of these analyses to the arsenic risk depending on the age or rice varieties are not well maintained. Please link these modellings to the health risk assessments.
Reviewer 3 Report
Not being a software/predictive tools expert myself, I find this manuscript very interesting and novel. As a toxicologist though, I can agree that contamination of foodstuff with heavy metals is unfortunately still a severe global problem.
I have only few specific comments towards your work from a toxicological point of view:
1. Materials and Methods -2.1 Data sources - since you already mention Chinese and USA national limits concerning the content of heavy metals in foodstuff, please also add recomendations of EFSA and EU limits (Regulation No 1881/2006) to get a more global and comprehensive overview.
2. Mercury ycles through the atmosphere, water and soil in various forms to different parts of the world. Due to this natural cycle, irrespective of which part of the world releases mercury it could affect an entirely different part of the world making mercury pollution a global concern. This needs to be mentioned in your study as this can affects the results of your predictions.
3.In a well lead discussion, also the drawbacks and week sides of your tool should be discussed. As bare minimum you should higlight the importance of monitoring (I mean actual laboratory analysis) that must be carried out to be able to ensure food quality and safety.
Reviewer 4 Report
The manuscript food-2109276 "Informer-based Safety Risk Prediction of Heavy Metals in Rice in China" deals with the topical issue of safety prediction and risk assessment for heavy metal contamination of rice.
The article is very interesting, but there are small errors:
1) For the phrase "Heavy metals" in the text, it is better to use the abbreviation TM.
2) Line 37: What half-life of Cd are the authors referring to? The main natural isotopes 112Cd and 114Cd are non-radioactive. Or are they other isotopes? Then it needs to be clarified.
3) Lines 96-99 and 134-136: What forms of heavy metals are we talking about - water-soluble, acid-soluble or gross? This needs to be clarified as different forms of metals have different toxicity.
4) Figures 11-15, in my opinion, should be moved to Supplementary and increased in size.
5) Line 459: "4. Discussion" correct to "4. Conclusion".
